# Single-Switch Non-Isolated Resonant DC-DC Converter for Single-Input Dual-Output Applications

Cristian Díaz-Martín [1], Eladio Durán [1,2,*], Salvador P. Litrán [1,3], José Luis Álvarez [1,4] and Jorge Semião [5,6]

1   Centro Científico-Tecnológico de Huelva (CCTH), University of Huelva, 21007 Huelva, Spain;
    cristian.diaz@diesia.uhu.es (C.D.-M.); salvador@uhu.es (S.P.L.); alvarez@dti.uhu.es (J.L.Á.)
2   Department of Electronic Engineering, Computer Systems and Automation, University of Huelva,
    21007 Huelva, Spain
3   Department of Electrical and Thermal Engineering, Design and Projects, University of Huelva,
    21007 Huelva, Spain
4   Department of Information Technologies, University of Huelva, 21007 Huelva, Spain
5   Institute Superior of Engineering (ISE), Universidade do Algarve, 8005-139 Faro, Portugal; jsemiao@ualg.pt
6   Instituto to de Engenharia de Sistemas e Computadores-Investigação e Desenvolvimento (INESC-ID),
    Universidade de Lisboa, 1000-029 Lisboa, Portugal
*   Correspondence: aranda@uhu.es; Tel.: +34-959217655

**Abstract:** This paper describes a new configuration of Cuk and SEPIC (Single-Ended Primary Converter) ZVS-QR (zero-voltage switching quasi-resonant) combination DC-DC converter for bipolar output with a single switch. The proposed topology employs a single ground-referenced power switch, which simplifies the gate drive design with a single *L-C* resonant network and provides a bipolar output voltage with good regulation, acceptable efficiency and a step-down/up conversion ratio. This configuration provides dual-output voltage by switching the power switch to zero voltage, which is an interesting alternative for many applications where small size, light weight and high power density are very important aspects. In order to verify its performance, a SEPIC–Cuk Combination ZVS-QR prototype with a cost-effective commercial resonant controller was designed and tested. The experimental results show that the proposed combined topology is suitable for Single-Input Dual-Output (SIDO) applications.

**Keywords:** bipolar output; DC-DC converters; resonant; single-input dual-output (SIDO); symmetric outputs





## 1. Introduction

In recent decades, the trends in power electronics have been related to reliability and high efficiency in high-power applications, while in low-power applications, the two major trends are related to high power density and low voltage; however, the main trend is related to miniaturization for integration. Small size and weight, together with high power density, are very important items in power supplies for small format applications, mobile electronic systems and onboard systems, driven by applications with severe restrictions on space installation, such as More-Electric Aircraft (MEA), Electric Vehicles (EVs), Robotics, Electric Ship (ES), Solid-State Lighting, Integrated DC-DC Power Supply and Monolithic Power Systems, Intelligent DC-DC Power Distribution (micro or nano grids), On-chip Power Supply, Wireless Remote-Sensing Node and Power Transfer, New Transport Technologies, Energy Harvesting and IoT.

Regarding power applications, DC-DC converters have a leading role and are widely used at all power levels. The main objective of DC-DC conversion is to transfer electric power from a source with one DC value to a load requiring a different DC value, which may be a low-loss conversion and, thus, require high efficiency. DC-DC switching converters have been very popular during the last three decades. They can decrease or increase the

magnitude of the DC voltage obtained at the output and/or reverse its polarity. Their function is similar to the one performed by a transformer in an AC-AC conversion.

A classification of the main switched DC-DC converters can be carried out according to the switching technique used: hard switching (PWM converters) and soft switching (resonant converters), with the resonant converters being the ones that have currently contributed the most to the reduction in size and weight. Converters based on soft-switching techniques use resonant *L-C* elements that cause the sinusoidal variation in voltage and current waveforms in order to reduce switching losses. This feature of the resonant DC-DC conversion techniques is used to shape the current and voltage waveforms of the switches to achieve either zero-voltage turn-on or zero-current turn-off. In fact, they were developed as an alternative to simple rectangular switching, and the power devices switch at high frequency, allowing the converter to maintain an acceptable efficiency (for light loads and full loads) by minimizing the significant switching losses associated with hard switching. In these converters, the size and weight are reduced (mainly magnetic and filtering components), and the power density is increased, with low stress on the switching devices. In addition, the soft-switching technique uses Variable Frequency (VF) for the regulation, with constant on-time or constant off-time. The *L-C* resonant elements generate an oscillating waveform that allows power devices' transitions to occur at zero voltage (zero-voltage switching, ZVS) or at zero current (zero-current switching, ZCS), eliminating or reducing switching losses in the converter and allowing higher-frequency operation, thus reducing the size of the magnetic and filtering components. Both resonant techniques (ZVS and ZCS) are widely used in many applications. Recent publications on ZVS and ZCS soft-switching converters can be found in [1–3]. Resonant converters can be classified from different points of view and topics: type of converter where they are applied (DC-DC, AC-DC, DC-AC, or AC-AC), applications, efficiency, operating frequency, conversion ratio, number of inputs and outputs, isolated and non-isolated versions, range of voltages, currents and powers [4–10]. However, the most extended classification is the one that considers the number of resonant circuits used and operation modes. In this sense, the converters are classified as: Resonant Power Converters (RPCs), Resonant Transition Converters (RTCs), Multi-Resonant Converters (MRCs) and Quasi-Resonant Converters (QRCs). RPCs are based on resonant tank stages with several reactive components [11], while RTCs consist of auxiliary switches as well as circuits with capacitors and inductors to produce the resonant action [12]. MRCs [13] and QRCs [14] also use resonant elements (*L* and *C*) to perform the resonant action, but with the following differences: In MRCs, the resonant action is performed on the two elements of the switching cell (the main switch and the freewheel diode); therefore, ZVS or ZCS conditions can be obtained over both devices. In QRCs, the resonant action is only performed on the main switch.

Generally, for any PWM converter, ZCS-QRC and ZVS-QRC configurations can be derived by adding a high-frequency *L-C* tank circuit to the main switching device. This way, resonant switching converters are an extension of hard-switching converters, but the switch is complemented with a high-frequency *L-C* tank circuit to provide resonant operation on the main switch, facilitate zero-current or zero-voltage switching and avoid simultaneous high current and high voltage in the switch device. This allows for quasi-resonant techniques and zero-voltage or zero-current switching on the main switch of DC-DC converters to be applied in traditional single-inductor converters, such as CSC (Canonical Switching Cell), Boost, Buck and Buck–Boost single-inductor converters, as well as in well-known two-inductor topologies such as Zeta, Cuk and SEPIC, in their different versions *L*-type and *M*-type (depending on how the *L-C* tank and switch are interconnected) and in half-wave and full-wave implementations (depending on whether the resonant action is performed over half cycle or full cycle), in order to obtain ZVS or ZCS conditions.

These configurations and topologies are well established in the literature and were developed several decades ago [15–17]. In the same way, the advantages and limitations of quasi-resonant converters are also well established. QRCs provide reduced EMI (radiated and conducted) due to the low $dv/dt$ and $di/dt$ produced (no higher peak currents),

sinusoidal waveforms that facilitate the turn-off process and voltage control by changing the switching frequency and zero current and/or zero voltage across the switches during switching transitions, thus substantially reducing switch losses. In addition, the parasitic components $L$ and $C$ can be incorporated as parts of the resonant circuit. Their main limitations are due to the fact that they require variable switching frequencies for regulation, which is very often considered a drawback of resonant converters. On-state currents and off-state voltages are higher than those found in non-resonant configurations of the same power.

In general terms, ZVS-QRCs are more widely used than ZCS-QRCs, mainly because of the parasitic junction capacitances associated with the semiconductor switch that can be incorporated as part of the required resonant capacitor. Also, in ZCS-QRCs, the charge stored in the parasitic capacitance associated with the semiconductor switch is dissipated in the switch itself during the conduction state. $L$-type and $M$-type configurations have similar behavior. The resonant capacitor in $M$-type configurations is exposed to a lower peak voltage, although it is bipolar. The main advantage of the ZCS-QR $L$-type converters is that, for some configurations, the resonant capacitor can be associated with the parasitic junction capacitance of the freewheel diode. To regulate the output voltage against input voltage and load variations, ZCS-QRCs require constant on-time control, while constant off-time control needs to be employed for ZVS-QRCs. With respect to half-wave and full-wave versions, for the same conditions, full-wave operation requires a lower switching frequency (higher switching period) and provides better load regulation than half-wave versions. ZCS-QRCs for half-wave operation mode require a unidirectional switch, while for full-wave operation, a bidirectional switch is required. In the same way, ZVS-QRCs require a bidirectional switch for half-wave operation and a unidirectional switch for full-wave operation. This causes the implementation of full-wave versions of ZVS-QRCs with MOSFET switches (and also ZCS-QRCs in half-wave versions) to require the use of an additional diode in series with the MOSFET to make them unidirectional, which is not very effective and increases the number of devices required. However, the advantages and disadvantages of each configuration and topology must be analyzed and compared individually for each specific case and application.

On the other hand, Single-Input Dual-Output (SIDO), bipolar symmetric outputs, dual and bipolar output DC-DC converters are receiving renewed interest in a wide variety of applications where two output voltages of the same magnitude but different signs are required from a single input supply. Among other reasons, this interest is due to the fact that SIDO converters can provide an additional higher supply voltage between both positive and negative output voltages. In this sense, it is increasingly common to find applications that require 24 V voltages from intermediate 12 V DC buses (Distributed Power Architecture, DPA), which in turn supply the traditional 1.8 V, 3.3 V and 5 V voltages. In this context, SIDO converters cover applications such as industrial, communications and computing power supplies [18], Electric Vehicles [19], LED lighting [20], as well as medical devices [21], photovoltaic systems [22], Low-voltage bipolar DC microgrids [23], advanced driver assistance (ADA) systems for cars, vans, trucks and buses (12 V and 24 V supplies) as emergency braking, detection systems (front and reverse cameras) for cyclists and pedestrians and intelligent speed assistance, which make driving more comfortable, safer and easier, and where small size and high power density are important aspects to account for. Many of the communications systems installed in commercial vehicles are powered by a 24 V battery and require up to 25 W of output power. A bipolar output is used to plug in electric vehicle charging stations in both AC and DC types and in DC fast charging, portable chargers, Grid-to-Vehicle (G2V) and Home-to-Vehicle (H2V), as well as Vehicle-to-Home (V2H), Vehicle-to-Grid (V2G), Vehicle-to-Vehicle (V2V), Vehicle-to-Building (V2B) and Vehicle-to-Everything (V2X) concepts.

In addition, the combination of basic converters can reduce the power density since it allows for the combination of basic configurations to obtain dual-output voltages with a single power switch and requires a smaller number of components. This converter

combination methodology has been proposed as a solution in hard switching [24] and soft switching [25], and it contemplates the combination of simple structures with the same front end, sharing switching nodes and giving rise to combined structures with different conversion ratios.

In this paper, a new combination of Cuk and SEPIC (Single-Ended Primary Converter) ZVS-QR converter for bipolar output is presented. This combined configuration produces a step-down/step-up-type converter that allows the development of dual-output transformerless converters. Two main additional advantages of the proposed combined configuration are highlighted: (i) simple structure, since the converter is designed with a single switch and does not require synchronization among the switches, which increases the power density; (ii) the control terminal of the switch is grounded, which simplifies the implementation of the gate drive.

In order to verify the proposed configuration, an experimental prototype has been developed with a cost-effective commercial resonant controller, and its performance has been confirmed for different regulating conditions. The prototype provides a dual-output voltage of $\pm12$ V from a single input in the range of 10–14 V, operating in step/up and step/down modes. The results obtained show that the designed converter is suitable for applications where SIDO voltage is required.

This work is organized as follows: In Section 2, the new topology is presented and analyzed, along with the operation mode. Section 3 describes the simulation model and the prototype developed, which have allowed, through experimental results, to verify the behavior of the proposed combination converter for several loads and operating modes. In Section 4, some conclusions are drawn.

## 2. Proposed Single-Switch SIDO ZVS-QRC Configuration Description

From conventional ZVS-QR $M$-type half-wave Cuk and SEPIC structures, shown, respectively, in Figures 1 and 2, it can be observed that, from a conceptual and analytical point of view, Cuk and SEPIC configurations are very similar and also present some of the advantages and limitations of resonant configurations described above. Both provide zero-voltage switching on the power switch ($Q$), use the same number of components, namely two inductors ($L_1$ and $L_2$), a freewheel diode ($D$) and a link capacitor ($C_1$) with the same resonant tank ($L_R$ and $C_R$); both also provide the same conversion ratio, but SEPIC is a positive output converter, while Cuk is a negative output converter; and both also have a single power switch connected to the negative terminal of the DC source ($V_g$).

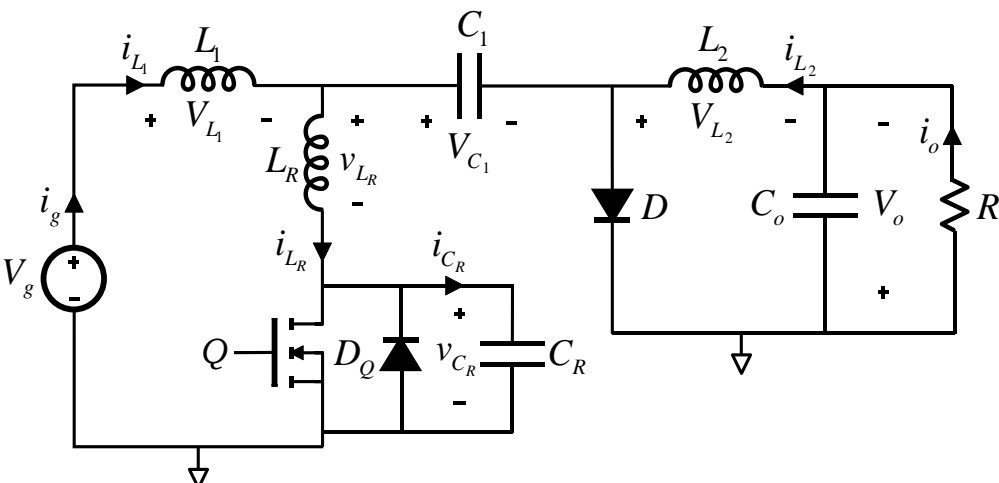

**Figure 1.** ZVS-QR $M$-type half-wave Cuk converter.

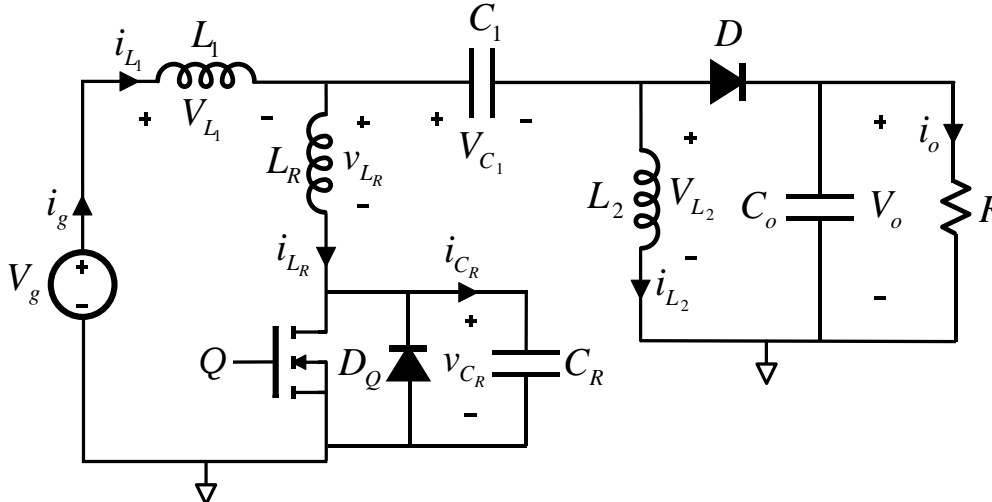

**Figure 2.** ZVS-QR *M*-type half-wave SEPIC converter.

If the configurations shown in Figures 1 and 2 are compared, it can be seen that both converters have an identical front part that can be combined as a common node and that it is possible to combine the two configurations to implement a SIDO converter, as shown in Figure 3. The combined structure shows a common $L_R$-$C_R$ tank, a common input inductor ($L_1$) and a single power switch ($Q$) connected to the negative terminal of the DC generator ($V_g$), and the resulting converter is able to provide a bipolar output voltage. This can be an interesting solution for many applications requiring small size and high power density. The main advantage of the proposed configuration is that it allows the implementation of bipolar symmetric outputs with only one controllable power switch and soft-switching techniques.

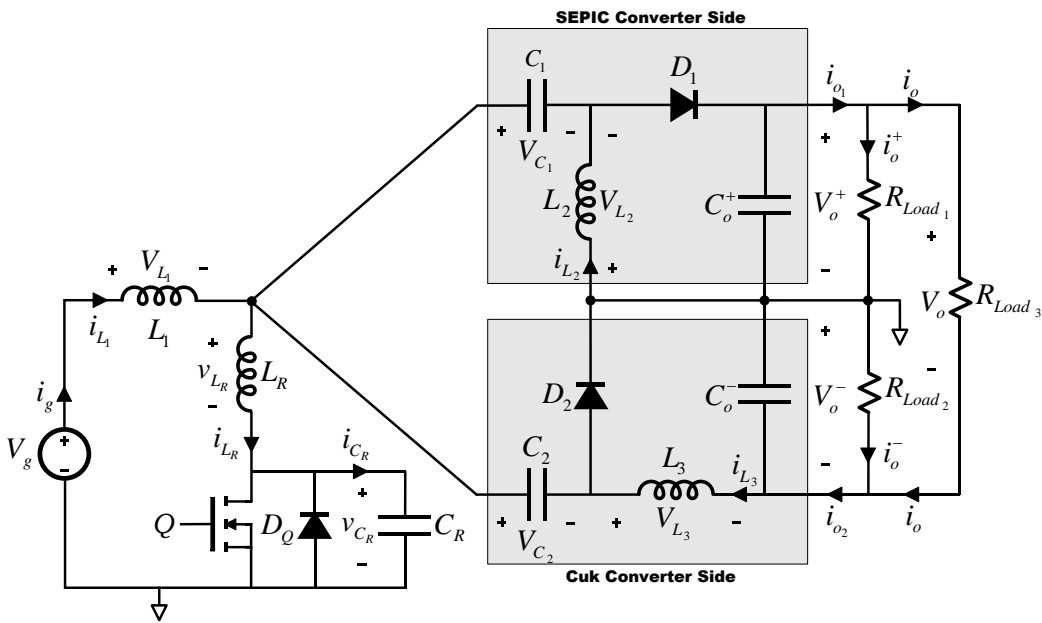

**Figure 3.** SIDO ZVS-QR SEPIC–Cuk combination converter.

### 2.1. Operation Mode

In a static way, the operation mode of the ZVS-QR SEPIC–Cuk combination converter shown in Figure 3 can be analyzed by assuming that the inductors' currents ($L_1$, $L_2$ and $L_3$) and the capacitors' voltages ($C_1$, $C_2$, $C_o{}^+$ and $C_o{}^-$) remain relatively constant during a switching period ($T_s$), close to their average values. In these conditions, the operation

mode can be divided into four time intervals during $T_s$. Therefore, it can be analyzed by the four equivalent circuits shown in Figure 4, assuming half-wave operation with respect to the resonant action ($D_Q$ conducts for a time interval and prevents the voltage across the resonant capacitor from changing its sign). The analysis begins with the capacitor $C_R$ voltage being zero, the load currents $I_{o1}$ and $I_{o2}$ circulating across the resonant inductor $L_R$ and the switch $Q$ being on.

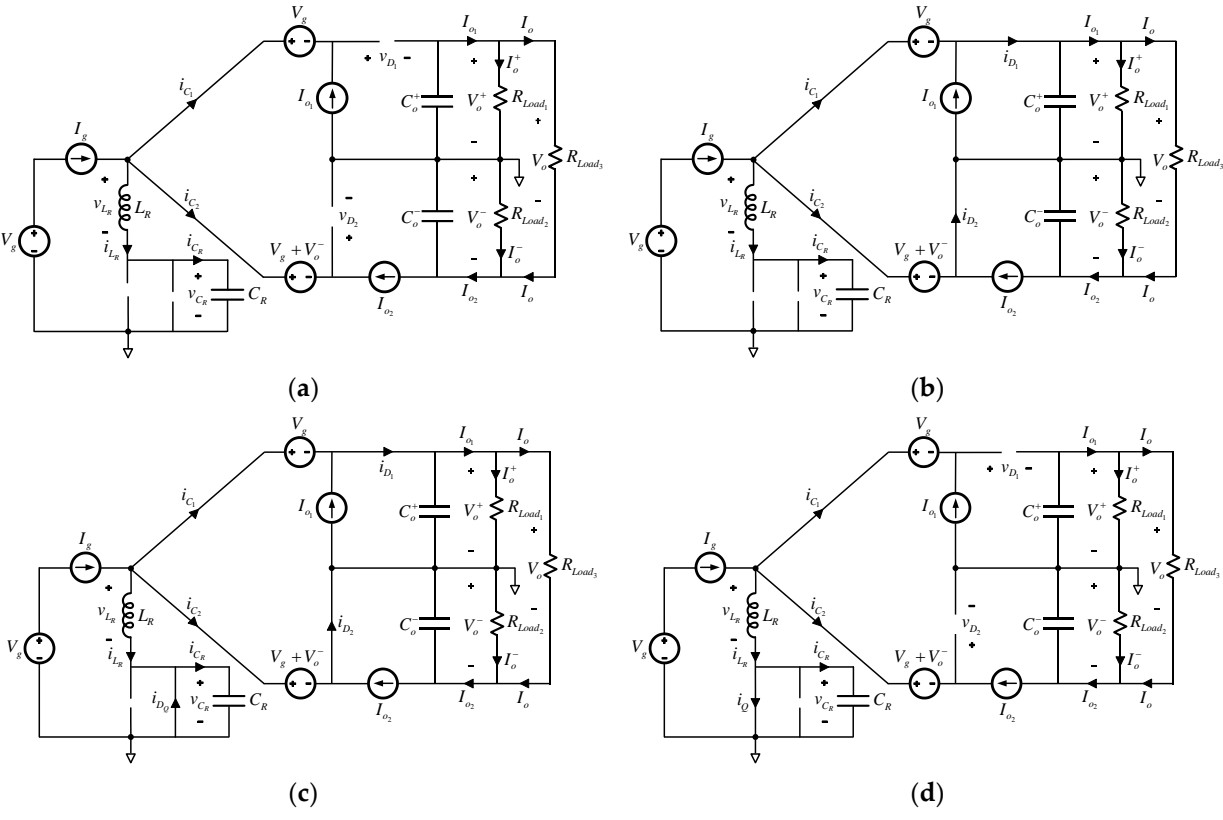

**Figure 4.** Equivalent circuits of the ZVS-QR SEPIC–Cuk combination converter during a switching period. (**a**) Linear State. Interval $[t_1 - t_0]$. (**b**) Resonant State. Interval $[t_2 - t_1]$. (**c**) Inductor-Discharging State. Interval $[t_3 - t_2]$. (**d**) Passive State. Interval $[t_4 - t_3]$.

Linear Interval $[t_1 - t_0]$:

At $t_0$, $Q$ is turned off. The supply current is provided from $V_g$ through $L_R$ and $C_R$. The voltage across $C_R$ (and across $Q$) grows linearly from zero (zero-voltage switching), and this time interval ends when the voltage across $C_R$ reaches $V_g + V_o/2$ and the freewheel diodes ($D_1$ and $D_2$) are directly biased (at time $t_1$).

Resonant Interval $[t_2 - t_1]$:

At $t_1$, the freewheel diodes $D_1$ and $D_2$ are turned on and conduct the current of the load, and that is due to $I_g$ and $i_{LR}$ (half for a balanced load). The voltage across $C_R$ varies sinusoidally, the inductor current $L_R$ varies cosinusoidally and the capacitor $C_R$ and the inductor $L_R$ are in resonance. The energy of the inductor is transferred to the capacitor, which increases its voltage from $V_g + V_o/2$ to the resonant wave amplitude. This time interval ends when the voltage across $C_R$ returns to zero and $D_Q$ is directly biased (at time $t_2$).

Inductor-Discharging Interval $[t_3 - t_2]$:

At $t_2$, $D_Q$ turns on. The voltage across $C_R$ attempts to reverse, but it is clamped to zero by $D_Q$, which is turned on. The inductor current grows linearly. $Q$ should be turned on before $D_Q$ is turned off. This time interval ends when $Q$ starts supplying the load currents ($I_{o1}$ and $I_{o2}$) again and the freewheel diodes ($D_1$ and $D_2$) are reverse-biased (at time $t_3$). Thus, $Q$ is turned on with zero-voltage switching.

Passive State Interval $[t_4 - t_3]$:

At $t_3$, $Q$ turns on and supplies the load currents ($I_{o1}$ and $I_{o2}$) again, and the conditions return to the initial ones. This time interval ends when $Q$ is turned off again, starting a new switching cycle. This time interval is also used to regulate the output voltages.

*2.2. Steady-State Analysis*

Assuming a ZVS-QR SEPIC–Cuk combination converter without losses (ideal components), the following assumptions can be considered:

(1)  The inductors $L_1$, $L_2$ and $L_3$ are considered current sources (currents are assumed to be constant and without any ripple), with $L_1$, $L_2$ and $L_3 \gg L_R$.

(2)  The link and output capacitors are considered voltage sources (voltages are assumed to be constant and without any ripple), with $C_1$, $C_2$, $C^+_o$ and $C^-_o \gg C_R$.

(3)  The $L_R$-$C_R$ tank is undamped (oscillatory).

(4)  The resonant wave amplitude is greater than $V_g + V_o/2$ ($V_m > V_g + V_o/2$).

The following parameters are also used in the analysis:

$$\text{Resonant angular frequency (rad/sec): } \omega_o = 2\pi f_o = \frac{1}{\sqrt{L_R \cdot C_R}}$$

$$\text{Characteristic impedance } (\Omega): Z_o = \sqrt{\frac{L_R}{C_R}} = \omega_o \cdot L_R = \frac{1}{\omega_o \cdot C_R}$$

$$\text{Equivalent Loads } (\Omega): \frac{1}{R_o^+} = \frac{1}{R_{Load_1}} + \frac{2}{R_{Load_3}} \tag{1}$$

$$\frac{1}{R_o^-} = \frac{1}{R_{Load_2}} + \frac{2}{R_{Load_3}}$$

$$\frac{1}{R_o} = \frac{1}{R_{Load_1}} + \frac{1}{R_{Load_2}} + \frac{4}{R_{Load_3}}$$

A switching period is divided into four intervals, as seen in Section 2.1. Considering the initial conditions of each interval, the differential equations can be solved with the help of the corresponding equivalent circuit (Figure 4) and the duration of the interval determined.

- $t_0 \le t \le t_1$; $Q$, $D_Q$, $D_1$ and $D_2$ off (Figure 4a).

The corresponding equations can be written as follows:

Initial Conditions:

$$i_{L_R}(t_0) = I_g + I_{o_1} + I_{o_2}, \ v_{L_R}(t_0) = 0, \ i_{C_R}(t_0) = I_g + I_{o_1} + I_{o_2} \text{ and } v_{C_R}(t_0) = 0$$

Differential Equations ($t_1 - t_0$):

$$i_{L_R}(t) = i_{C_R}(t) \text{ and } V_g + \frac{V_o}{2} = v_{C_R}(t) + v_{L_R}(t) \tag{2}$$

$$\Rightarrow v_{C_R}(t) = \frac{1}{C_R} \int_{t_0}^{t_1} i_{L_R}(t) \, dt \Rightarrow \frac{(I_g + I_{o_1} + I_{o_2})}{C_R} \cdot (t_1 - t_0) = V_g + \frac{V_o}{2}$$

Interval Duration:

$$t_1 - t_0 = C_R \frac{\left(V_g + \frac{V_o}{2}\right)}{\left(I_g + I_{o_1} + I_{o_2}\right)}$$

The freewheel diodes' ($D_1$ and $D_2$) voltages, which are related to the output voltages, are given by

$$v_{D_1}(t) = V_g + V_o^+ - v_{C_R}(t) = \left(V_g + V_o^+\right)\left(1 - \frac{t - t_0}{t_1 - t_0}\right)$$

$$v_{D_2}(t) = V_g + V_o^- - v_{C_R}(t) = \left(V_g + V_o^-\right)\left(1 - \frac{t - t_0}{t_1 - t_0}\right) \tag{3}$$



- $t_1 \leq t \leq t_2$; $Q$ and $D_Q$ off, $D_1$ and $D_2$ on (Figure 4b).

  Initial Conditions:

  $i_{L_R}(t_1) = I_g + I_{o_1} + I_{o_2}$, $v_{L_R}(t_1) = 0 \Rightarrow \frac{di_{L_R}}{dt}(t_1) = 0$, $v_{C_R}(t_1) = V_g + \frac{V_o}{2}$

  and $i_{C_R}(t_1) = I_g + I_{o_1} + I_{o_2} \Rightarrow \frac{dv_{C_R}}{dt}(t_1) = \frac{I_g + I_{o_1} + I_{o_2}}{C_R}$

  Differential Equations $(t_2 - t_1)$:

  $i_{L_R}(t) = i_{C_R}(t)$ and $V_g + \frac{V_o}{2} = v_{C_R}(t) + v_{L_R}(t)$

  $\Rightarrow V_g + \frac{V_o}{2} = v_{C_R}(t) + L_R \frac{di_{L_R}}{dt} \xrightarrow{\frac{d}{dt}} 0 = \frac{i_{L_R}(t)}{C_R} + L_R \frac{d^2 i_{L_R}}{dt^2}$

  $\Rightarrow i_{L_R}(t) = (I_g + I_{o_1} + I_{o_2}) cos\omega_o(t - t_1)$

  and $v_{C_R}(t) = V_g + \frac{V_o}{2} + (I_g + I_{o_1} + I_{o_2}) \cdot Z_o sin\omega_o(t - t_1)$

  while: $i_{D_1}(t) = (I_g - i_{L_R}(t))\left(\frac{R_o}{R_o^+}\right) + I_o^+$

  and $i_{D_2}(t) = (I_g - i_{L_R}(t))\left(\frac{R_o}{R_o^-}\right) + I_o^-$

  (4)

  Interval Duration:

  $$t_2 - t_1 = \frac{1}{\omega_o}\left(sin^{-1}\left(\frac{(V_g + \frac{V_o}{2})}{(I_g + I_{o_1} + I_{o_2}) \cdot Z_o}\right) + \pi\right)$$

- $t_2 \leq t \leq t_3$; $Q$ or $D_Q$ on, $D_1$ and $D_2$ on (Figure 4c).

  Initial Conditions:

  $i_{L_R}(t_2) = (I_g + I_{o_1} + I_{o_2}) cos\omega_o(t_2 - t_1)$

  $v_{L_R}(t_2) = V_g + \frac{V_o}{2}$, $i_{C_R}(t_2) = 0$ and $v_{C_R}(t_2) = 0$

  Differential Equations $(t_3 - t_2)$:

  $v_{L_R}(t) = V_g + \frac{V_o}{2}$ and $v_{C_R}(t) = 0$

  $\Rightarrow i_{L_R}(t) = \frac{1}{L_R}\int_{t_2}^{t_3} v_{L_R}(t)\, dt + i_{L_R}(t_2)$

  $i_{L_R}(t_3 - t_2) = \frac{1}{L_R}\left(V_g + \frac{V_o}{2}\right)(t_3 - t_2) + (I_g + I_{o_1} + I_{o_2}) cos\omega_o(t_2 - t_1)$

  while: $i_{D_1}(t) = (I_g - i_{L_R}(t))\left(\frac{R_o}{R_o^+}\right) + I_o^+$

  and $i_{D_2}(t) = (I_g - i_{L_R}(t))\left(\frac{R_o}{R_o^-}\right) + I_o^-$

  (5)

  Interval Duration:

  $$t_3 - t_2 = L_R \frac{(I_g + I_{o_1} + I_{o_2})}{(V_g + \frac{V_o}{2})}[1 - cos\omega_o(t_2 - t_1)]$$

- $t_3 \leq t \leq t_4$; $Q$ on, $D_Q$, $D_1$ and $D_2$ off (Figure 4d).

  Conditions:

  $i_{L_R}(t) = I_g + I_{o_1} + I_{o_2}$, $v_{L_R}(t) = 0$, $i_{C_R}(t) = 0$ and $v_{C_R}(t) = 0$

  (6)

The waveforms of the ZVS-QR SEPIC–Cuk combination converter during a switching period and the four intervals analyzed are shown in Figure 5.

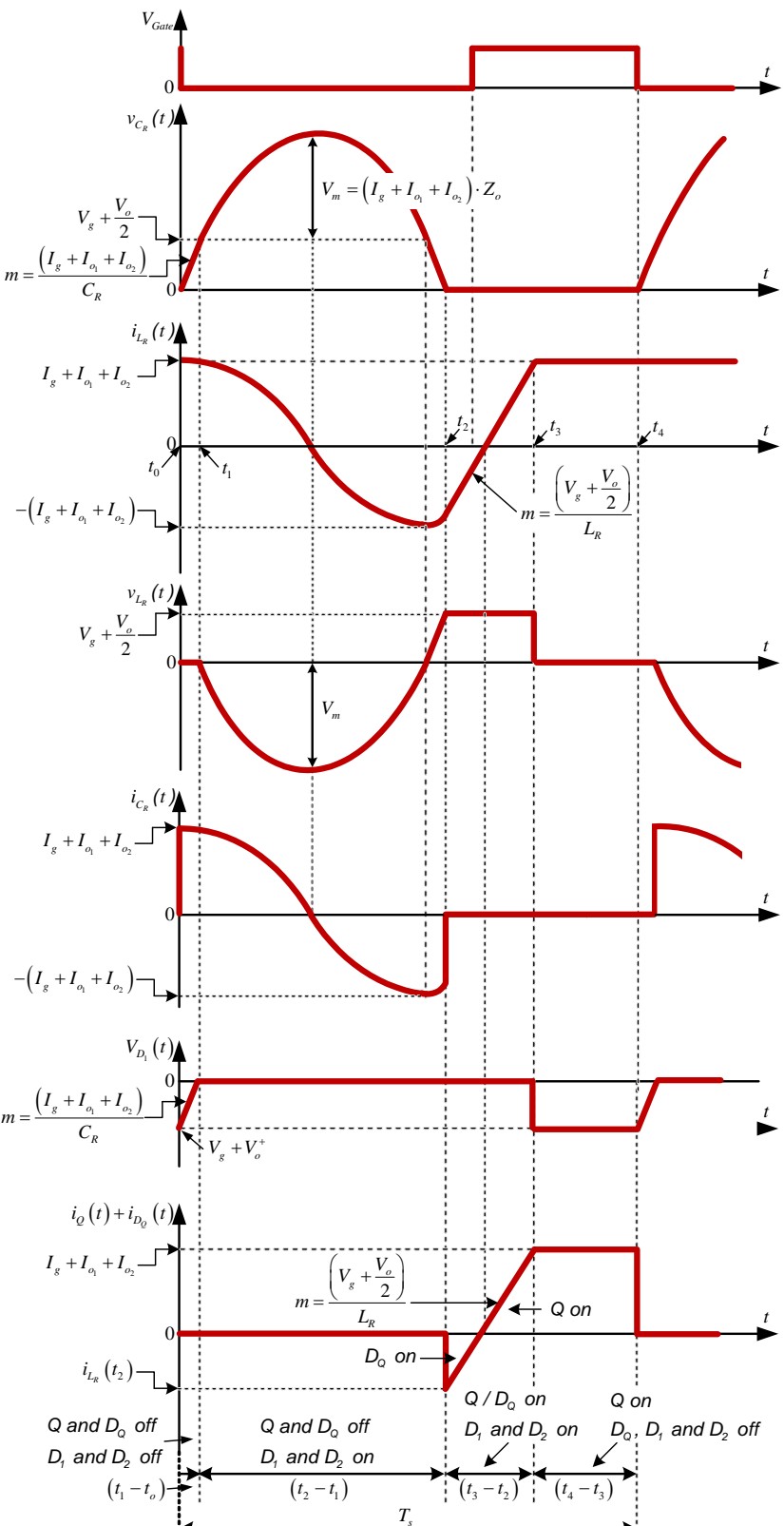

**Figure 5.** Main waveforms of the ZVS-QR SEPIC–Cuk combination converter during a switching period.

The output voltages can be evaluated by considering the average voltage across the freewheel diodes ($D_1$ and $D_2$) for a switching period (waveforms shown in Figure 5), since these voltages are filtered by the output capacitors.

The current conversion relation can be expressed by

$$I_g = (I_{o_1} + I_{o_2}) \cdot \frac{\left(1 - f_s \cdot \left(t_3 - \frac{t_1}{2}\right)\right)}{f_s \cdot \left(t_3 - \frac{t_1}{2}\right)} = (I_o^+ + I_o^- + 2I_o) \cdot \frac{\left(1 - f_s \cdot \left(t_3 - \frac{t_1}{2}\right)\right)}{f_s \cdot \left(t_3 - \frac{t_1}{2}\right)} \quad (7)$$

Conversion relations can also be expressed by defining the parameter $m$ as:

$$V_o^+ = V_o^- = V_g \cdot \frac{(1-m)}{m} \Rightarrow V_o = 2V_g \cdot \frac{(1-m)}{m}$$

$$I_g = (I_{o_1} + I_{o_2}) \cdot \frac{(1-m)}{m} = (I_o^+ + I_o^- + 2I_o) \cdot \frac{(1-m)}{m} \quad (8)$$

With $m = f_S \cdot \left(t_3 - \frac{t_1}{2}\right)$; $0 < m < 1$

Output Voltages:

$$V_{D_1}(t) \text{ and } V_{D_2}(t) = \begin{cases} \left[\left(V_g + \frac{V_o}{2}\right) - \frac{(I_g + I_{o_1} + I_{o_2})}{C_R}(t - t_1)\right] & t_o \leq t \leq t_1 \\ 0 & t_1 \leq t \leq t_2 \\ 0 & t_2 \leq t \leq t_3 \\ \left(V_g + \frac{V_o}{2}\right) & t_3 \leq t \leq t_4 \end{cases}$$

$$\left.\begin{array}{l} V_o^+ = V_{D_1}(t)_{AVG} \\ V_o^- = |V_{D_2}(t)_{AVG}| \end{array}\right\}$$

$$V_o^+ = V_o^- = \frac{1}{T_s}\int_{t_o}^{t_1} \left[\left(V_g + \frac{V_o}{2}\right) - \frac{(I_g + I_{o_1} + I_{o_2})}{C_R}(t - t_1)\right] dt + \frac{1}{T_s}\int_{t_3}^{t_4} \left(V_g + \frac{V_o}{2}\right) dt$$

Hence

$$V_o^+ = V_o^- = \left(V_g + \frac{V_o}{2}\right)\left[1 - \frac{1}{T_s}\left(t_3 - \frac{t_1}{2}\right)\right] \Rightarrow V_o^+ = V_o^- = V_g \cdot \frac{\left(1 - f_s \cdot \left(t_3 - \frac{t_1}{2}\right)\right)}{f_s \cdot \left(t_3 - \frac{t_1}{2}\right)}$$

$$\Rightarrow V_o = 2V_g \cdot \frac{\left(1 - f_s \cdot \left(t_3 - \frac{t_1}{2}\right)\right)}{f_s \cdot \left(t_3 - \frac{t_1}{2}\right)} \quad (9)$$

Table 1 summarizes the voltages and currents for the main components of the bipolar output ZVS-QR SEPIC–Cuk combination converter in terms of input voltage ($V_g$), the conversion parameter $m$ and loads $R_{Load_1}$, $R_{Load_2}$ and $R_{Load_3}$.

From the above analysis, the following characteristics of the bipolar output ZVS-QR SEPIC–Cuk combination converter can be highlighted:

(1) Both outputs provide a step-down/step-up conversion ratio.
(2) The turn-on and turn-off transitions of the main power switch occur at zero voltage, so switching losses are low.
(3) Output voltages depend on load currents.
(4) The $L_R$-$C_R$ tank must be dimensioned so that the zero-switching condition is satisfied:
$$\left(\left(V_g + \frac{V_o}{2}\right) < (I_g + I_{o_1} + I_{o_2}) \cdot Z_o\right).$$

**Table 1.** Voltages and Currents for the Main Components of the ZVS-QR SEPIC–Cuk Converter.

| | Switch $Q$ | $D_1$ | $D_2$ |
|---|---|---|---|
| Peak Voltage on Semiconductors | $\frac{V_g}{m}\left(1+\frac{(1-m)\cdot Z_o}{R_o\cdot m}\right)$ | $\frac{V_g}{m}$ | $\frac{V_g}{m}$ |
| Average Current across the Semiconductors | $\frac{V_g\cdot(1-m)^2}{R_o\cdot m^2}$ | $\frac{V_g\cdot(1-m)}{R_o^+\cdot m}$ | $\frac{V_g\cdot(1-m)}{R_o^-\cdot m}$ |
| | $V_{C1}$ | $V_{C2}$ | |
| Average Voltage across the Link Capacitors | $V_g$ | $\frac{V_g}{m}$ | |
| | $I_{L1,AVG}$ | $I_{L2,AVG}$ | $I_{L3,AVG}$ |
| Average Current across the Filter Inductors | $\frac{V_g\cdot(1-m)^2}{R_o\cdot m^2}$ | $\frac{V_g\cdot(1-m)}{R_o^+\cdot m}$ | $\frac{V_g\cdot(1-m)}{R_o^-\cdot m}$ |
| | **Average** | **Peak** | |
| Voltage across the Resonant Capacitor ($C_R$) | $V_g$ | $\frac{V_g}{m}\left(1+\frac{(1-m)\cdot Z_o}{R_o\cdot m}\right)$ | |
| | **Average** | **Peak** | |
| Current across the Resonant Inductor ($L_R$) | $\frac{V_g\cdot(1-m)^2}{R_o\cdot m^2}$ | $\frac{V_g\cdot(1-m)}{R_o\cdot m^2}$ | |

$$With:\ \frac{1}{R_o^+}=\frac{1}{R_{Load_1}}+\frac{2}{R_{Load_3}}$$
$$\frac{1}{R_o^-}=\frac{1}{R_{Load_2}}+\frac{2}{R_{Load_3}}$$
$$\frac{1}{R_o}=\frac{1}{R_{Load_1}}+\frac{1}{R_{Load_2}}+\frac{4}{R_{Load_3}}$$
$$m=f_S\cdot\left(t_3-\frac{t_1}{2}\right);\ 0<m<1$$

Nomenclature: $V_g$: source voltage; $m$: conversion parameter; $R_o$: equivalent load resistance; $Z_o$: characteristic impedance; $t_1$: end of linear state interval; $t_3$: end of inductor-discharging state interval; $f_s$: switching frequency.

## 3. Model and Experimental Prototype

In order to verify the analysis of the SIDO ZVS-QR SEPIC–Cuk combined converter, simulation and experimental results were carried out using a simulation model and an experimental prototype for low-power but scalable for higher-power values. Therefore, the circuit in Figure 3 has been implemented and developed as shown in Figure 6. The objective is to provide results that will allow the validation of the newly developed configuration.

In a similar way as in the design of single-input single-output converters, in the ZVS-QR SEPIC–Cuk combination converter, some basic rules must be considered. The values of the inductances $L_1$, $L_2$ and $L_3$ must be chosen in such a way that they can limit the ripple in the input current ($\Delta i_{L1}$) and the output currents ($\Delta i_{L2}$ and $\Delta i_{L3}$). In this sense, the following inequalities must be satisfied:

$$\Delta i_{L_1}=\frac{V_g}{L_1}\left(T_s-t_{off}\right)<<I_g$$
$$\Delta i_{L_2}=\frac{V_o^+}{L_1}t_{off}<<I_{o_1}$$
$$\Delta i_{L_3}=\frac{V_o^-}{L_3}t_{off}<<I_{o_2}$$

(10)

Similarly, for the selection of the output filtering capacitors, the maximum acceptable ripple of the converter output voltage must be considered. In this sense, the voltage ripples in $C^+_o$ and $C^-_o$ are given by:

$$\frac{\Delta v_o^+}{V_o^+}=\frac{\left(T_s-t_{off}\right)}{R_o^+\cdot C_o^+},\text{for SEPIC side}$$
$$\frac{\Delta v_o^-}{V_o^-}=\frac{t_{off}}{8\cdot L_3\cdot C_o^-\cdot f_s},\text{for Cuk side}$$

(11)

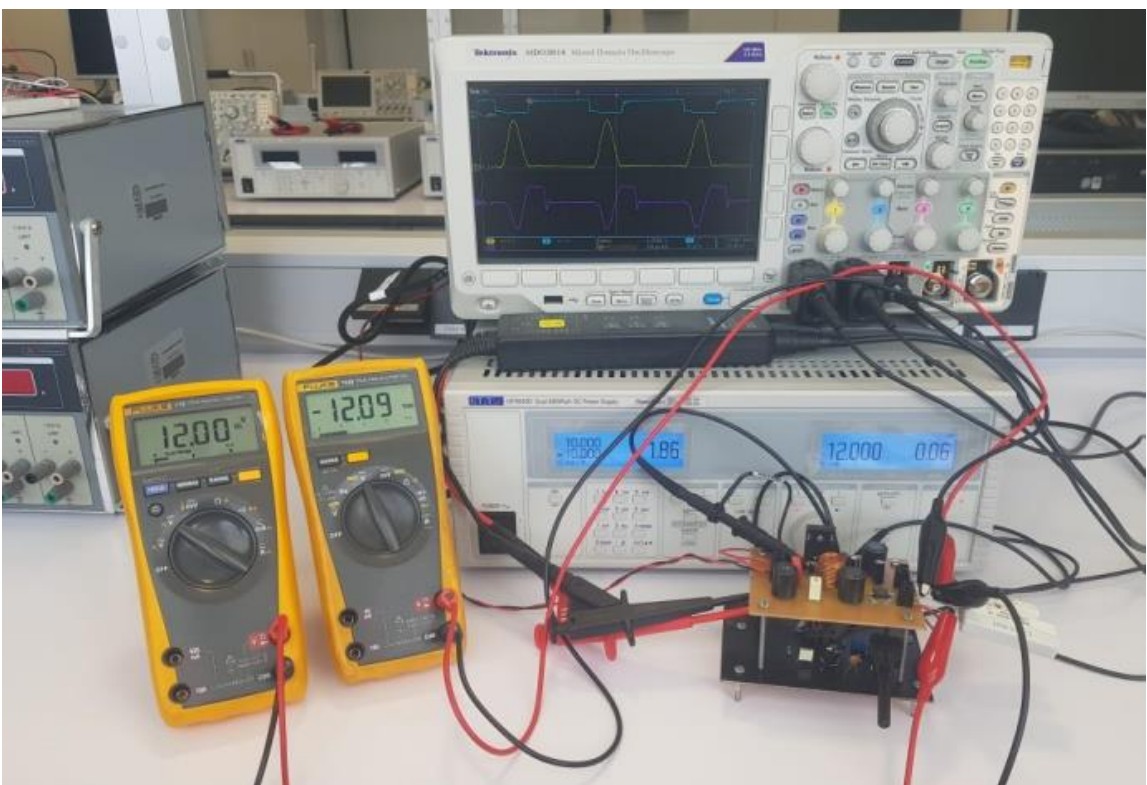

**Figure 6.** Experimental prototype.

In addition, the selection of the link capacitors is based on the assumption that the voltage in those capacitors must be constant. Their values must be chosen so that the resonant frequency, formed by inductor filters and link capacitors, is much lower than the switching frequency. So, for Cuk and SEPIC, the resonance frequency is given by:

$$\omega_{r_1}^2 = \frac{1}{C_1\,(L_1+L_2)}\textit{for SEPIC side}$$

$$\omega_{r_2}^2 = \frac{1}{C_2(L_1+L_3)}\textit{for Cuk side}$$

*furthermore*:

$$C_1 > \frac{\left(T_s-t_{off}\right)^2}{2\cdot T_s\cdot R_o^+} \text{ and } C_2 > \frac{\left(T_s-t_{off}\right)^2}{2\cdot T_s\cdot R_o^-}$$

(12)

In this regard, the voltage ripple in $C_1$ and $C_2$ is given by:

$$\frac{\Delta v_o^+}{V_o^+} = \frac{\left(T_s-t_{off}\right)}{R_o^+\cdot C_o^+}\textit{,for SEPIC side}$$

$$\frac{\Delta v_o^-}{V_o^-} = \frac{t_{off}}{8\cdot L_2\cdot C_o^-\cdot f_s}\textit{,for Cuk side}$$

(13)

while for the selection of $L_R$ and $C_R$ (resonant tanks), we have considered the conversion ratio based on the parameter *m* defined in Table 1. For an input voltage range of 10 to 14 V, *m* varies between 0.45 and 0.54. For *m* = 0.54, the switching frequency is established around 535 kHz, and the time instants $t_1$ and $t_3$ are determined by (7)–(9). This allows the values of $L_R$ and $C_R$ to also be determined, considering the power (7 W for each output) and establishing a commercial standard starting value for the resonant inductor.

The proposed converter components and specifications are given in Table 2.

**Table 2.** Specifications and Components Used for the SIDO ZVS-QR SEPIC–Cuk Combination Converter.

| | |
|---|---|
| Input Voltage ($V_g$) | 10–14 V |
| Output Voltage ($V_o{}^+$, $V_o{}^-$) | $\pm$12 V |
| Switching Frequency ($f_s$) | 550–180 kHz |
| Power | 30 W |
| Switch ($Q$) | IRFP1405 (65 V, 95 A, 5.3 m$\Omega$) |
| Freewheel Diodes ($D_1$, $D_2$) | MBR360 (60 V, 3 A, 0.73 V at 3 A, 25 °C) |
| Filter Inductors ($L_1$, $L_2$, $L_3$) | 220 $\mu$H, 1.5 A |
| Link Capacitors ($C_1$, $C_2$) | 1 $\mu$F, 16 V |
| Output Capacitors ($C_o{}^+$, $C_o{}^-$) | 330 $\mu$F, 16 V |
| Resonant Inductor ($L_R$) | 2.1 $\mu$H |
| Resonant Capacitor ($C_R$) | 10 nF, 16 V |

### 3.1. Simulation Results

The simulation conditions have been established to obtain two bipolar voltages of $\pm$12 V from an input voltage source ($V_g$) of 10 and 14 V. For the power switch, a MOSFET model was used. The trigger pulses are generated through a pulse generator designed to maintain the average value of the output voltage at $\pm$12 V. The switching frequency is set to 411 kHz when $V_g$ is 10 V and to 541 kHz when $V_g$ is 14 V, with $L_R$ = 2.1 $\mu$H and $C_R$ = 12 nF, which establishes an off-time of 900 ns. The values chosen for the rest of the passive components in the converter are: $L_1 = L_2 = L_3$ = 220 $\mu$H, $C_1 = C_2$ = 1 $\mu$F and $C^+{}_o = C^-{}_o$ = 330 $\mu$F.

The ZVS-QR SEPIC–Cuk combination converter has been analyzed using a simulation model, first in step-down mode. The waveforms shown in Figure 7 correspond to a simulation with $R_{Load_1}$ and $R_{Load_2}$, first with 10 $\Omega$ and then with 20 $\Omega$ (where the resonant action is maintained in both cases with $V_g$ = 14 V). This has allowed us to check the intervals' duration, the peak and average values of the resonant waveforms, the current and the voltage, provided by (2)–(6), for the specified load range and the following variables: voltage in the gate terminal ($V_{Gate}$), resonant inductor current ($i_{L_R}$), resonant capacitor voltage ($v_{C_R}$), resonant inductor voltage ($v_{L_R}$), resonant capacitor current ($i_{C_R}$), freewheel diodes voltage ($V_{D_1}$ and $V_{D_2}$) and current through the MOSFET ($i_Q$).

In the same way, the results shown in Figure 8 correspond to $V_g$ = 14 V, $R_{Load_1}$ and $R_{Load_2}$ of 10 $\Omega$, but with a change in the switching frequency from 411 kHz to 200 kHz at 0.075 s. In this case, due to the change in frequency, the output voltages vary between $\pm$11.8 and $\pm$22.7 V, and there is a change in the converter operation from step-down to step-up mode. In addition, the results shown in Figure 9 also correspond to $V_g$ = 14 V, with a switching frequency of 411 kHz, but with a change in $R_{Load_1}$ and $R_{Load_2}$ from 20 $\Omega$ to 10 $\Omega$ also at 0.075 s. As a result, the output voltages vary between $\pm$11.7 and $\pm$17 V, and the converter also changes operating mode (from step-up to step-down), but now due to the change in the load.

In the same way, the simulation results shown in Figure 10 correspond to the behavior of the combined converter in step-up mode, with $R_{Load_1}$ and $R_{Load_2}$ first of 10 $\Omega$ and then 20 $\Omega$ (where the resonant action is maintained in both cases) and $V_g$ = 10 V. These results have made it possible to verify the performance of the ZVS-QR SEPIC–Cuk combination converter shown in Table 1. As it can be seen, all the waveforms agree with the analytical analysis, which has been shown in Figure 5. In addition, the conversion ratio is also satisfied quite accurately. Each waveform is analyzed with the analytically obtained values in order to verify the consistency of the proposed converter.

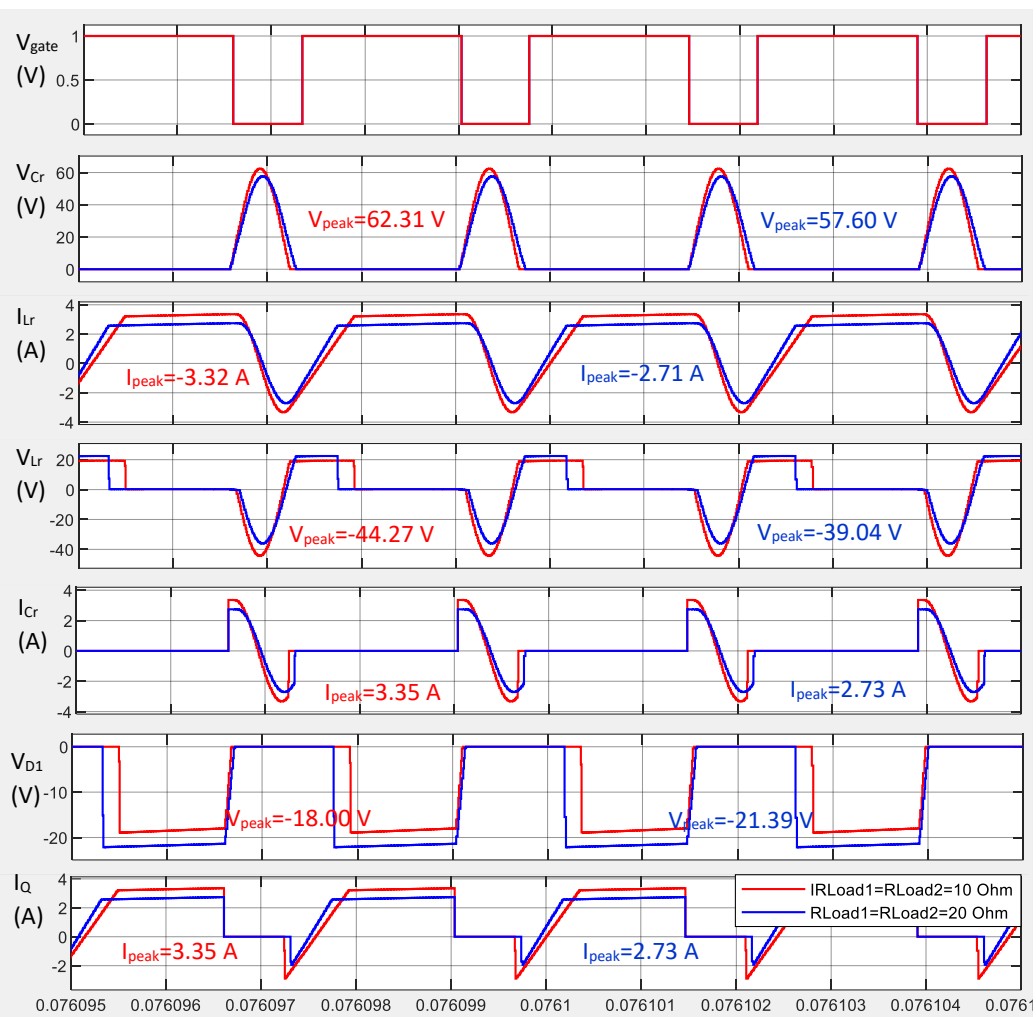

**Figure 7.** Waveforms of the ZVS-QR SEPIC–Cuk converter in step-down mode.

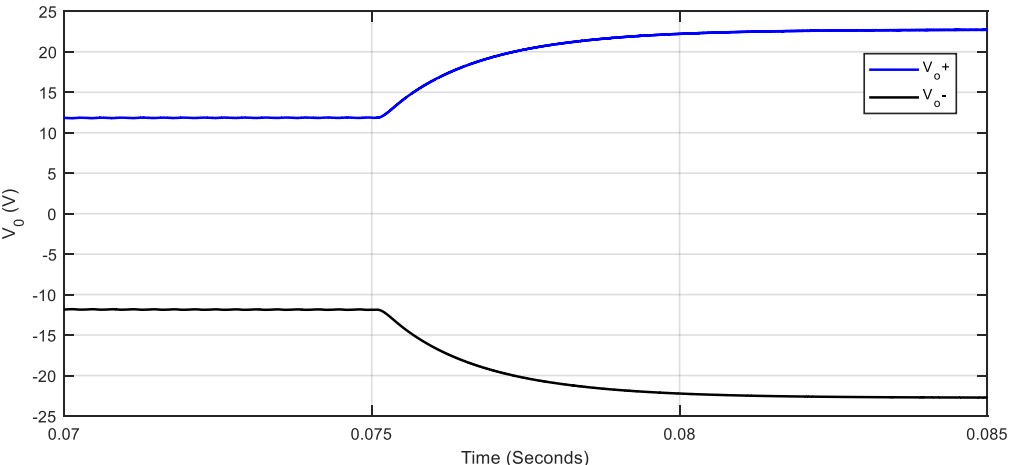

**Figure 8.** Output voltage variation with $R_{Load_1}$ and $R_{Load_2}$ 10 Ω, $V_g$ = 14 V, with a change in the switching frequency from 411 kHz to 200 kHz at 0.075 s.

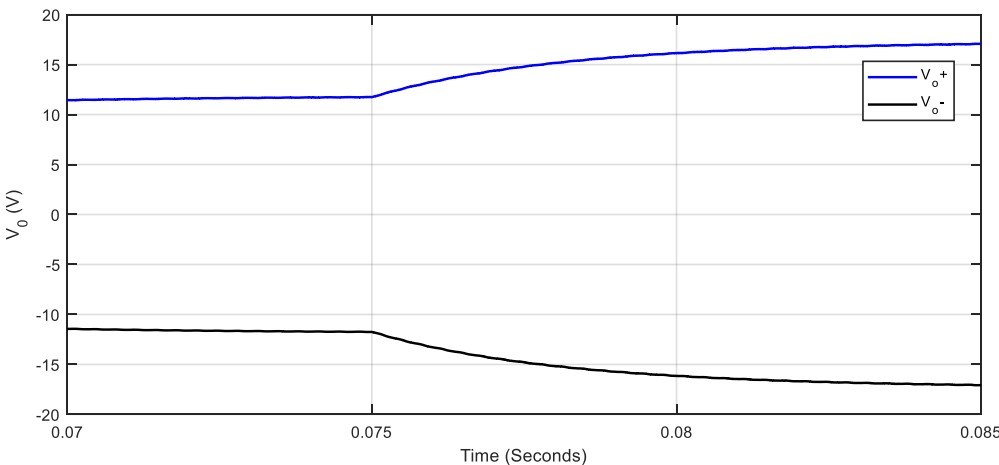

**Figure 9.** Output voltage variation at switching frequency from 411 kHz, $V_g$ = 14 V, with a change in $R_{Load_1}$ and $R_{Load_2}$ from 20 Ω to 10 Ω at 0.075 s.

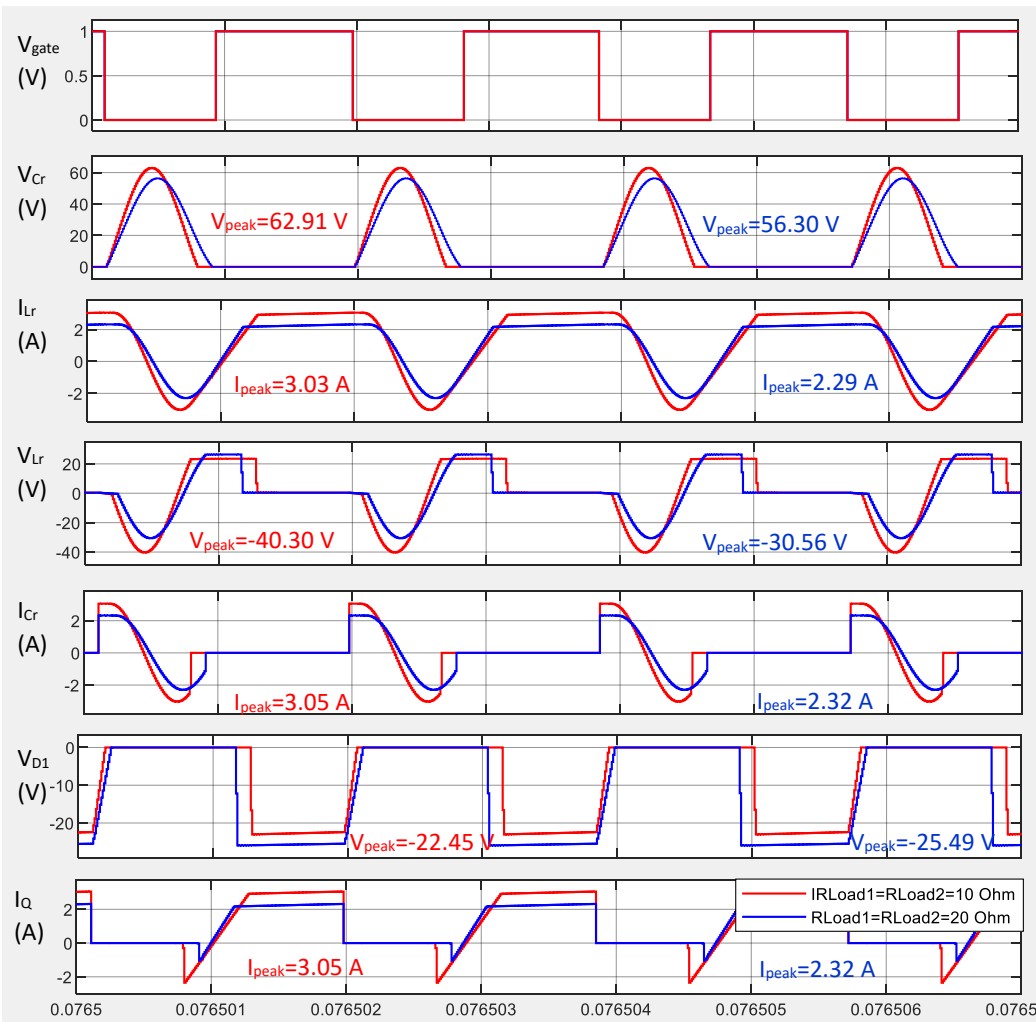

**Figure 10.** Waveforms of the ZVS-QR SEPIC–Cuk converter in step-up mode.

On the other hand, the results shown in Figure 11 correspond to $V_g$ = 10 V, $R_{Load_1}$ and $R_{Load_2}$ of 10 Ω, but with a change in the switching frequency from 200 kHz to 290 kHz at 0.075 s. In this case, due to the change in frequency, the output voltages vary between ±11.9 and ±16.2 V, and the converter remains in step-up mode. In addition, the results shown in Figure 12 also correspond to $V_g$ = 10 V, with a switching frequency of 290 kHz, but with a change in $R_{Load_1}$ and $R_{Load_2}$ from 20 Ω to 10 Ω also at 0.075 s. In this case, the output voltages vary between ±12.1 and ±16.4 V, and the converter does not change its operating mode either, remaining in step-up.

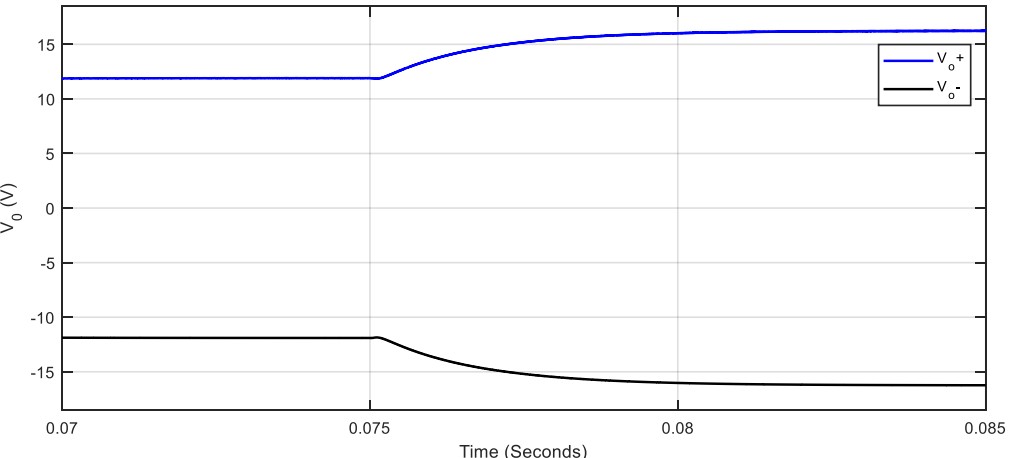

**Figure 11.** Output voltage variation with $R_{Load_1}$ and $R_{Load_2}$ of 10 Ω, $V_g$ = 10 V, with a change in switching frequency from 200 kHz to 290 kHz at 0.075 s.

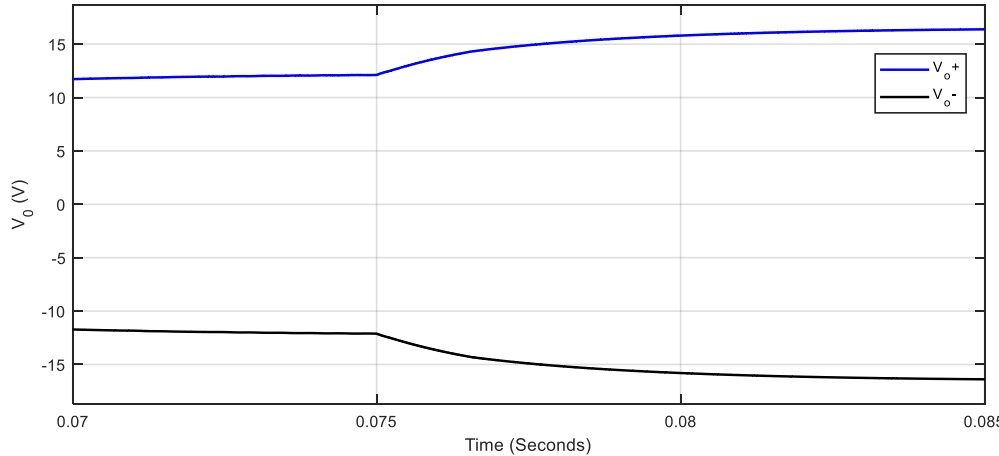

**Figure 12.** Output voltage variation at switching frequency from 290 kHz, $V_g$ = 10 V, with a change in $R_{Load_1}$ and $R_{Load_2}$ from 20 Ω to 10 Ω at 0.075 s.

### 3.2. Experimental Results

The experimental prototype developed (Figure 6) allows the performance of the converter to be verified under different test conditions and also in step-down and step-up operation modes.

The block diagram of the laboratory experimental prototype converter is shown in Figure 13. It consists mainly of a SIDO ZVS-QR SEPIC–Cuk combination converter, a resonant controller and a gate-driver circuit.

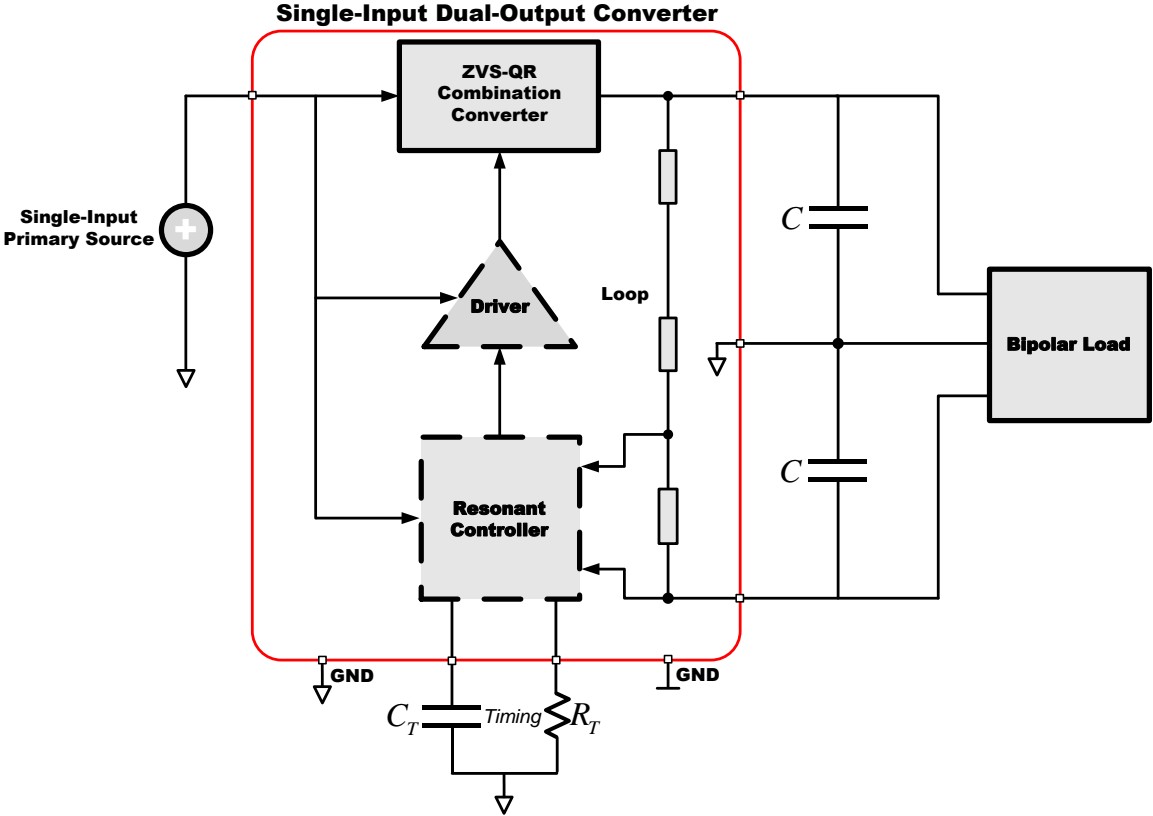

**Figure 13.** Block diagram of the developed prototype.

The control scheme is developed by a cost-effective commercial resonant-mode controller (UC3863). This controller is optimized for zero-voltage switching quasi-resonant converters. Mainly, it consists of an internal voltage-controlled oscillator (VCO) and an error amplifier. It is used in this application to regulate the output voltages to $\pm$12 V from an input voltage ranging from 10 V to 14 V. The UC3863 allows setting a minimum and maximum frequency between 10 kHz and 1 MHz. The controller is configured to generate a minimum of 380 kHz and a maximum of 540 kHz switching frequencies for ZVS operation in step-up and step-down modes.

The control signal generated by the controller needs to be adapted to an adequate level for the switching transistor gate. The gate-driver circuit must provide sufficient output current to charge the gate capacity within the required time. In this sense, the UCC27423 commercial gate driver used ensures proper switching conditions.

The results shown in Figures 14 and 15 correspond to two different situations: when $V_g$ is 10 V and when $V_g$ is 14 V, both step-up and step-down configurations, with two balanced loads of 600 mA, one at the positive output and the other at the negative output ($R_{Load_1}$ and $R_{Load_2}$, respectively). Some measurements were made with multimeters that, in the range of 0–1 kV DC, have a maximum resolution of 0.1 mV and an accuracy of 0.15%. The waveforms were acquired by an analog four-channel oscilloscope with an analog bandwidth of 1 GHz and an analog sample rate of 5 GS/s. For voltage measurements, a differential voltage probe was used; its bandwidth was 200 MHz, the input impedance was 5 MΩ and the maximum common-mode voltage was $\pm$750 V. An AC/DC current measurement probe, which provides a bandwidth greater than 120 MHz with selectable 5 A and 30 A measurement ranges, was also used; it also provides low-current measurement capability and accuracy to current levels as low as 1 mA, and it measures current pulses up to 50 A.

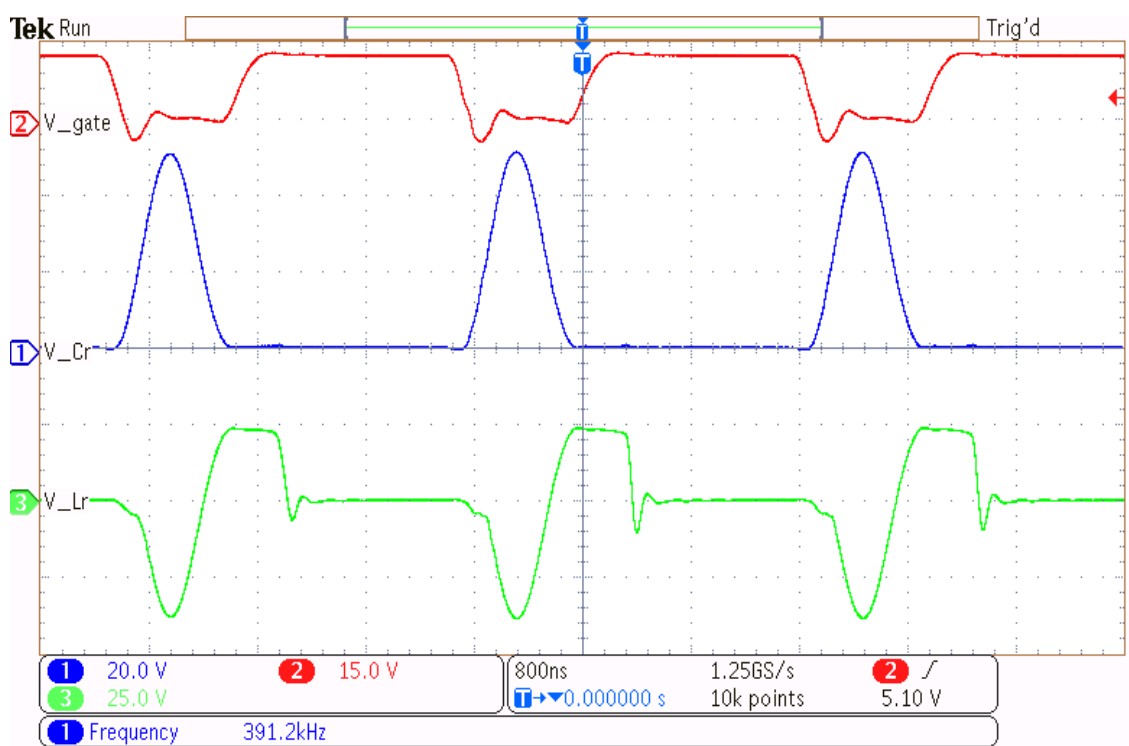

**Figure 14.** ZVS operation and step-up mode experimental results: Channel C1: 20 V/div, 0 V offset; Channel C2: 15 V/div, 45 V offset; Channel C3: 25 V/div, −50 V offset.

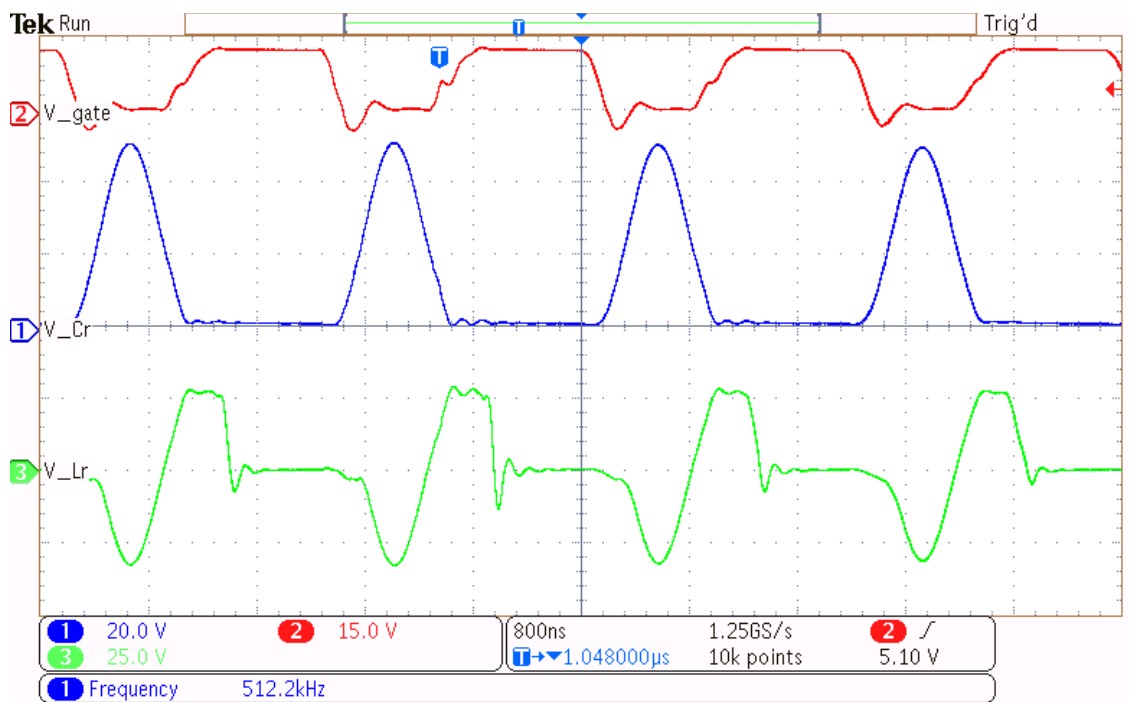

**Figure 15.** ZVS operation and step-down mode experimental results: Channel C1: 20 V/div, 0 V offset; Channel C2: 15 V/div, 45 V offset; Channel C3: 25 V/div, −50 V offset.

In step-up mode, the input voltage is set to 10 V to obtain two dual outputs of ±12 V. For this, a switching frequency of 390 kHz with an off-time of 848 ns is obtained with a load connected to the positive output and another equal load connected to the negative output (two monopolar loads). While in step-down mode, the input voltage is set to 14 V to obtain two dual outputs of ±12 V; in this case, a switching frequency of 530 kHz with

an off-time of 645 ns is obtained. Again, a load is connected to the positive output, and another equal one is connected to the negative output (two monopolar loads). In the same way, Figure 16 shows the input and output voltages for the input voltages of 10 V (on the left) and 14 V (on the right). The load is composed of two resistors of 20 Ω connected to the positive and negative outputs. It can be seen that the designed control allows for an output voltage of 12 V with a small variation around ±0.4 V. In these test conditions (and power used), in both step-down and step-up modes, the measured and obtained efficiency was 86%, which is a reasonable value for many applications. In this regard, Table 3 compares the proposed converter with some recently developed bipolar output DC-DC converters shown in References [18–22], taking into account characteristics such as the number of switches, diodes, capacitors and inductors used in the converter with bipolar output voltage capability, voltage stress, input/output conversion ability, use of an isolation transformer, power and efficiency tested, switching frequency, floating switch/es and soft-switching technique used.

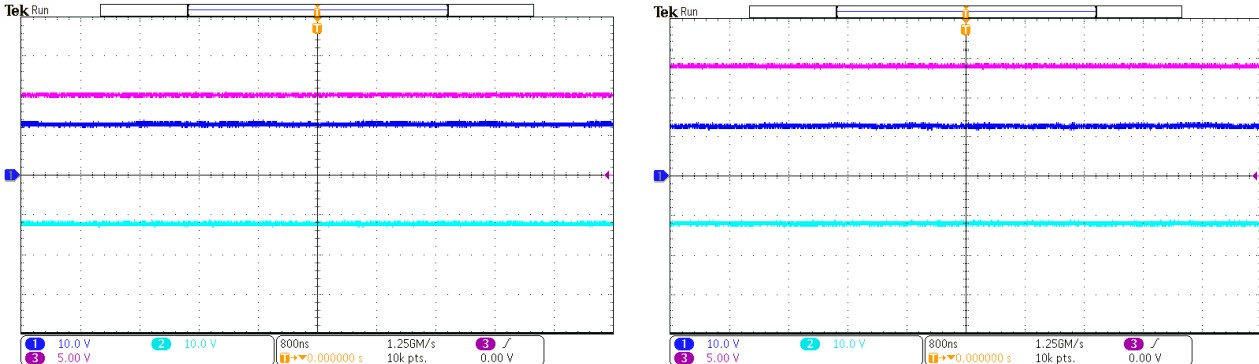

**Figure 16.** ZVS-QR SEPIC–Cuk converter in step-down and step-up modes for an input voltage of 10 V (on the **left**) and 14 V (on the **right**): SEPIC-side output voltage (Channel C1: 10 V/div, 0 V offset), Cuk-side output voltage (Channel C2: 10 V/div, 0 V offset) and input voltage (Channel C3: 5 V/div, 0 V offset).

**Table 3.** Comparison characteristics of the ZVS-QR SEPIC–Cuk converter with different converters with bipolar output voltage capability.

| Characteristics | [18] | [19] | [20] | [21] | [22] | Prop. Converter |
|---|---|---|---|---|---|---|
| Number of Switches | 2 | 4 | 4 | 4 | 1 | 1 |
| Number of Diodes | 2 | - | 1 | - | 2 | 2 |
| Number of Capacitors | 4 | 5 | 4 | 2 | 4 | 5 |
| Number of Inductors | 2 | 2 | 1 | 1 | 3 | 3 |
| Voltage Stress | $V_g$ | $V_g$ | $V_0$ | $V_o$ | $V_g + V_o$ | $> V_g + V_o$ |
| Conversion Ratio | Step-Down | Step-Down | Step-up | Step-up | Step-up Step-Down | Step-up Step-Down |
| Transformer | No | Yes | No | No | No | No |
| Floating Switch/es | Yes | Yes | Yes | Yes | No | No |
| Soft-Switching | ZVS | Yes | No | No | No | Yes |
| Power | 60 W | 3.3 kW | 3.5 W | 200 mW, | 1 kW | 30 W |
| Efficiency | 94.6% | 95% | 89.3% | 83.4% | 84.4% | 86% |
| Switching Frequency | 1 MHz | 190 kHz | 1 MHz | 2.5 MHz | 25 kHz | 550 kHz |

As a result, although the proposed converter provides a lower efficiency and operates at an acceptable frequency when compared to the other topologies in the literature, it requires a single non-floating power switch and two freewheel diodes and provides a bipolar output voltage with a step-down/step-up conversion ratio.

## 4. Conclusions

In this paper, a new configuration of the Cuk and SEPIC ZVS-QR combination DC-DC converter for bipolar output with a single switch has been presented. The topology proposed employs a single power switch with ground reference, which simplifies the design of the gate drive with a single resonant *L-C* circuit and provides a bipolar output voltage with good regulation, acceptable efficiency and a step-down/step-up conversion ratio. This combined configuration is an interesting alternative in many applications where small size, light weight and high power density are very important features. To verify its performance, a SEPIC–Cuk Combination ZVS prototype with a cost-effective commercial resonant-mode controller and gate driver was used. The prototype provides a dual-output voltage of ±12 V from a single input in the range 10–14 V, operating in step/up and step/down modes. The experimental results show that this combination is suitable for applications where a Single-Input Dual-Output (SIDO) voltage is required. Four aspects of the prototype proposed can be highlighted: (i) simple structure, since it uses only one switch and fewer passive elements; (ii) the driver and the control circuit are simpler structures, due to the existence of only one power switch to be controlled; (iii) a bipolar output voltage is provided without a transformer; and (iv) the turn-off and turn-on processes of the power switch occur at zero voltage, which reduces the switching losses.

**Author Contributions:** Investigation, C.D.-M., E.D., S.P.L., J.L.Á. and J.S. All authors have read and agreed to the published version of the manuscript.

**Funding:** This research received no external funding.

**Institutional Review Board Statement:** Not applicable.

**Informed Consent Statement:** Not applicable.

**Data Availability Statement:** The study did not report any data.

**Conflicts of Interest:** The authors declare no conflict of interest.

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
