# Peer review of "Single-Switch Non-Isolated Resonant DC-DC Converter for Single-Input Dual-Output Applications"

_applsci, doi:10.3390/app13158798_

Round 1

Reviewer 1 Report

Please provide state-space model for further controller design

Please provide model with both resistive load and constant power load

Please compare switch stress, efficiency, gain etc. with other advanced topologies

Please comment on control

please improve grammer

Author Response

The Authors are grateful for reviewers’ comments. Changes have been included to improve the paper according to his suggestions. We have rewritten some parts of the paper and corrected some errors, with the aim of improving paper.

Please provide state-space model for further controller design

Authors’ replay:

We are grateful for reviewer`s consideration.

The choice of how to manage the control of the proposed SEPIC-Cuk SIDO ZVS-QR combined converter was not a simple decision. This aspect of the submitted work has been extensively discussed and analysed by the authors, in terms of interest for the paper. But once we decided to implement a prototype of the combined converter, we considered different types of controllers: a control based on a classical proportional-integral controller, analogy and digitally controller, current mode and voltage mode, and others such as peak current mode control, valley current mode control, average current mode control and hysteretic control. But not all of them support a variable frequency and constant off-time control, required by the ZVS-QR converters. Furthermore, the use of any of these controllers, as the reviewer rightly proposes, would require providing a model of the controller used, which, in turn, would focus the paper in a different direction from the one intended, which is to present a new combined converter with dual output transformerless and a single switch. For these reasons, the authors have decided to opt for a commercial controller such as the UC3863. This device is optimized for the control of zero voltage switching quasi-resonant converters, and it is used in different applications and with various resonant converters topologies.

According to the reviewer`s recommendation an extended information on UC3863 controller has been added (please see page 17 of the revised paper).

“The control scheme is developed by a cost-effective commercial Resonant-Mode Controller, UC3863. This controller is optimized for zero voltage switching quasi resonant converters. Mainly, it consists of an internal Voltage Controlled Oscillator (VCO) and an error amplifier. It is used in this application to regulate the output voltages to ± 12 V, from an input voltage ranging from 10 V to 14 V. The UC3863 allows setting a minimum and maximum frequency between 10 kHz to 1 MHz. The controller is configured to generate a minimum of 380 kHz and a maximum of 540 kHz switching frequency for ZVS operation in step-up and step-down modes.”

Please provide model with both resistive load and constant power load

Authors’ replay:

We are grateful for reviewer`s consideration.

From our understanding, constant power adjusts the load current inversely with the load voltage so that the load power is constant, P = VI. It is this inverse property of a constant power load that is often useful in stability analysis. In this regard, it is difficult to test the proposed resonant combined converter with a constant power load. Since one of the characteristics of resonant converters is their sensitivity to load current changes and their low regulation, since a change in such current does not guarantee the resonant action and therefore, neither the output voltage regulation. The main applications of resonant converters are in systems with constant or almost constant load.

On the other hand, we have provided a model valid during the four time-intervals that lasts for one switching period, which has allowed us to describe the operation mode and the perform analysis, and that is consistent with simulation and experimental results (please see Figure 4 of the revised paper).

Please compare switch stress, efficiency, gain etc. with other advanced topologies

Authors’ replay:

According to the reviewer`s comments a new table has been added to compare the proposed converter and some recently developed bipolar output DC-DC converters shown in the References, taking into account characteristics such as: Number of Switches, Diodes, Capacitors and Inductors used in the converter with bipolar output voltage capability, voltage stress, input/output relation conversion ability, use of isolation transformer, power and efficiency tested, and switching frequency, floating switch/es and soft-switching technique used (please see Table 3 of the revised paper).

Please comment on control

Authors’ replay:

We have added a text on the commercial controller used, which tries to describe the control system used (please see page 17 of the revised paper).

Please improve grammer

We have revised and rewritten some parts of the paper and corrected some errors, with the aim of improving paper. In revised version, all changes have been highlighted in yellow.

Reviewer 2 Report

The paper is well written and interesting to read.

Minor grammar and syntax issues need correction.

Results must be developed.

more simulation results and formal comparison of results are needed.

I propose adding a list of abbreviations and symbols.

The conclusion isn’t accuracy. 

Minor editing of English language required

Author Response

The Authors are grateful for reviewers’ comments. Changes have been included to improve the paper according to his suggestions. We have rewritten some parts of the paper and corrected some errors, with the aim of improving paper.

The paper is well written and interesting to read.

Authors’ replay:

We are grateful for reviewer`s consideration.

Minor grammar and syntax issues need correction.

Authors’ replay:

We have revised and rewritten some parts of the paper and corrected some errors grammar and syntax issues, with the aim of improving paper. In revised version, all changes have been highlighted in yellow.

Results must be developed. More simulation results and formal comparison of results are needed.

Authors’ replay:

Four new Figures (Figures 8, 9, 11 and 12 of the revised paper) have been added with an explanation of the results shown in these figures (please see new Figures 8, 9, 11 and 12 and Section 3 of the revised paper).

I propose adding a list of abbreviations and symbols.

Authors’ replay:

According to the reviewer's proposed, a list of nomenclature has been included in Table 1. The variables used in each formula are defined according with the configuration presented (please see new Table 1 of the revised paper).

Reviewer 3 Report

1. Please tell me, are there plans to work on a converter with more power than 15W?

2. Please tell me, what were the power-losses in the semiconductor elements?

3. Please explain the measurement system, i.e. the list of measurement elements, calculation of measurement uncertainties, etc.

.

Author Response

The Authors are grateful for reviewers’ comments. Changes have been included to improve the paper according to his suggestions. We have rewritten some parts of the paper and corrected some errors, with the aim of improving paper.

  1. Please tell me, are there plans to work on a converter with more power than 15W?

Authors’ replay:

We are grateful for reviewer`s consideration.

Yes, we agree. This aspect has also been discussed by the authors, in terms of interest for the paper and for the readers. However, at this time, we have developed and evaluated a prototype with a power of 30W (15W for each output) and since the results have been satisfactory, we have decided to prepare a proposal for publication.

  1. Please tell me, what were the power-losses in the semiconductor elements?

Authors’ replay:

We are grateful for reviewer`s consideration.

In this topology power losses occur in the passive and active elements (power switch and diodes). Regarding losses in the passive elements, these are mainly produced in the inductances and its value depends on the quality factor of the coils, which depends on the design criterion for the same. On the other hand, the power losses in the switch are associated with resistance when the switch is on and with the diode conduction voltage. According to the design criteria adopted in the converter these losses can be around 60% of total.

  1. Please explain the measurement system, i.e. the list of measurement elements, calculation of measurement uncertainties, etc.:

Authors’ replay:

According to the reviewer's recommendation, we have added a new paragraph which tries to summarize the elements and equipment used to perform the measurements provided in the experimental results (please see page 18 of the revised paper).

“Some measurements were made with multimeters that, in the range of 0-1 kV DC, have a maximum resolution of 0.1 mV and an accuracy of 0.15%. The waveforms were acquired by an Analog four-channel oscilloscope, with an Analog Bandwidth of 1 GHz and an Analog Sample Rate of 5GS/s. For voltage measurements, a differential voltage probe was used and its bandwidth is 200 MHz, the input impedance 5 MΩ, and the maximum common mode voltage is ±750 V. It was also used an AC/DC current measurement probe, which provides a bandwidth greater than 120 MHz, with selectable 5 A and 30 A measurement ranges; it also provides low-current measurement capability and accuracy to current levels as low as 1 mA, and it measures current pulses up to 50 A peak.”

Reviewer 4 Report

The manuscript describes a new configuration of Cuk and SEPIC (Single Ended Primary Converter) ZVS-QR (Zero Voltage Switching-Quasi Resonant) combination DC-DC converter for bipolar output with single-switch. Although the paper is suitable to be published in the Journal, the following next points that should be improved are:

1.- Introduction Section need to be completed with more details about the advantages (and, of course, the possible drawbacks) of the proposed configuration with regards to the classic Cuk and SEPIC topologies... What are the advantages of the proposal? Please, clarify these advantages and, of course, the possible drawbacks of the manuscript proposal.

2.- Although the results presented in the paper are suitable, notice that results included in Figs. 7 and 8 (Waveforms of the ZVS-QR SEPIC-Cuk converter in step-down & step-up moded) are poor. Please, improve the quality of these two figures.

3.- In parallel, regarding the previous point, please, improve the explanation regarding the results presented in the paper.

4.- Finally, Conclusion Section should also be improved to highlight the advantages of the proposal.

Author Response

The Authors are grateful for reviewers’ comments. Changes have been included to improve the paper according to his suggestions. We have rewritten some parts of the paper and corrected some errors, with the aim of improving paper.

The manuscript describes a new configuration of Cuk and SEPIC (Single Ended Primary Converter) ZVS-QR (Zero Voltage Switching-Quasi Resonant) combination DC-DC converter for bipolar output with single-switch. Although the paper is suitable to be published in the Journal, the following next points that should be improved are:

 Authors’ replay:

 We are grateful for reviewer`s consideration.

Introduction Section need to be completed with more details about the advantages (and, of course, the possible drawbacks) of the proposed configuration with regards to the classic Cuk and SEPIC topologies... What are the advantages of the proposal? Please, clarify these advantages and, of course, the possible drawbacks of the manuscript proposal.

Authors’ replay:

According to the reviewer’s recommendation, we have revised and rewritten Introduction Section and also Section 2 Proposed Single-Switch SIDO ZVS-QRC Configuration Description, trying to provide more details on the advantages and disadvantages of the resonant configurations in general, and of the proposed configuration in particular (please see pages 2, 3 and 4, and Sections 1 and 2 of the revised paper).

Although the results presented in the paper are suitable, notice that results included in Figs. 7 and 8 (Waveforms of the ZVS-QR SEPIC-Cuk converter in step-down & step-up moded) are poor. Please, improve the quality of these two figures.

Authors’ replay:

According to the reviewer’s recommendation, we have changed Figures 7 and 8 to improve their quality (please see new Figures 7 and 10 of the revised paper).

In parallel, regarding the previous point, please, improve the explanation regarding the results presented in the paper.

Authors’ replay:

According to the reviewer’s recommendation, four new figures (Figures 8, 9, 11 and 12 of the revised paper) have been added, together with an explanation of the results shown in these figures, please see new Figures 8, 9, 11 and 12 and Section 3 of the revised paper.

Finally, Conclusion Section should also be improved to highlight the advantages of the proposal.

Authors’ replay:

According to the reviewer’s recommendation, we have revised and rewritten Conclusion Section, trying to highlight the advantages of the proposal (please see Conclusion Section of the revised paper).

Round 2

Reviewer 1 Report

Please add experimental results for load and input change

na

Author Response

The Authors are grateful for reviewers’ comments. Changes have been included to improve the paper according to his suggestions. 

Please add experimental results for load and input change

Authors’ replay:

The authors are grateful for the reviewer's comments. New experimental results have been added. A new Figure 16 has been included showing the input and output voltage waveforms when for two different input voltages of 10 V and 14 V. Also a new text has been added.

Reviewer 3 Report

.

Author Response

The Authors are grateful for reviewer’s comments.

Round 3

Reviewer 1 Report

Authors responded to comments

fine